# Lithium Occurrence in Italy—An Overview

**Andrea Dini [1], Pierfranco Lattanzi [2], Giovanni Ruggieri [2],[*] and Eugenio Trumpy [1]**

[1] Pisa Headquarters, CNR, Istituto di Geoscienze e Georisorse, 56124 Pisa, Italy; andrea.dini@igg.cnr.it (A.D.); eugenio.trumpy@igg.cnr.it (E.T.)

[2] Florence Territorial Unit, CNR, Istituto di Geoscienze e Georisorse, 50121 Florence, Italy; pierfrancolattanzi@gmail.com

[*] Correspondence: giovanni.ruggieri@igg.cnr.it

**Abstract:** Italy has no record of Li production, even though it is well known for its outstanding Li mineral specimens from the Elba Island pegmatites. Because of the current geopolitical situation, the opportunity for a systematic appraisal of resources is evident. Most European Li production comes from deposits associated with Late Paleozoic magmatic rocks. In Italy, such rocks occur extensively in Sardinia and Calabria, but their potential for Li is unknown, and deserves a more systematic exploration. Also of potential interest are the Permo–Triassic spodumene pegmatites in the Austroalpine units of the Central Alps. The Tertiary pegmatites (Elba Island and Central Alps) contain Li minerals, but do not appear large enough to warrant bulk mining. However, we notice that Tertiary–Quaternary magmatic rocks of the Tuscan and Roman magmatic provinces have systematically higher Li contents than those recorded in normal arc igneous rocks worldwide. Specifically, Tuscan granites contain up to 350 µg/g Li, mostly hosted by biotite (up to 4000 µg/g Li); the Capo Bianco aplite (Elba Island) contains up to 1000 µg/g. There are other small Li occurrences associated with Mn deposits and metabauxites, and there is a hypothetical potential for sediment-hosted deposits in the post-orogenic Lower Permian Alpine basins. However, the most promising potential seems to be associated with subsurface fluids. High-enthalpy fluids in geothermal fields may contain up to 480 mg/L Li. Lower-temperature thermal waters may also contain significant Li (>10 mg/L). Moreover, a visionary, but not impossible, perspective may consider a deep injection of water to interact with, and extract Li from, magmatic rocks.

**Keywords:** lithium; Italy; geothermal fluids

## 1. Introduction

Lithium appeared only recently in the history of humankind, having been discovered by J.A. Arfwedson in 1817 [1]. For a long time, its uses were restricted to specialized fields such as high-temperature lubricants, some types of ceramics and glass, and anti-depressive drugs [2]. In the last decade, the lithium market was revolutionized by the booming demand for rechargeable batteries, currently representing more than half of global lithium use [2]. By 2030, the global lithium demand is expected to be nearly 1.8 million tonnes (Mt) of lithium carbonate equivalent (LCE), i.e., almost six times the demand in 2020 [3]. Such a dramatic increase has fostered worldwide efforts for the exploration of new resources, as well as the reconsideration of previously known deposits.

The currently exploited lithium resources in the world belong to two types of deposits: brines and hard rock ores [4]. More than 60% of global lithium production occurs in brines, while lithium ores account for the rest. Continental brine deposits occur in high-elevation, arid regions (Andes, North American Cordillera, and Tibetan Plateau), where they are hosted by recent basins containing lacustrine evaporates [4]. The availability of easily leach-able volcanic rocks in the catchment basin or buried at depth is a ubiquitous characteristic. Recently, the possibility of exploiting unconventional brines (deep groundwater brines and geothermal fluids) has received considerable attention [5–7].

Hard-rock lithium ores are dominated by the so-called rare element granitic pegmatites, usually associated with S-type granite suites of highly variable age, e.g., [8]. World-class deposits with spodumene content up to 50 wt% are exploited in Australia, Canada, the USA, Brazil, and Zimbabwe. In addition to the classic spodumene ores, exploration has also been focused on pegmatites with a high content of lepidolite and lithium phosphates (e.g., montebrasite and lithiophilite), and on their zinnwaldite-rich, metasomatic contact zones [9].

Other hard-rock Li ores are represented by volcanic clay deposits (Mexico, USA, Peru; [4,5]) where lithium is either structurally bonded in smectites and micas (e.g., hectorite and tainiolite) or adsorbed as ion on clays (illite–smectite). A possible extension of this class of deposits is the unique deposit of Jadar in Serbia [10]. The deposit occurs in Miocene intramontane lacustrine sediments; the metal is hosted by a borosilicate of lithium and sodium, named jadarite [11], supposedly formed via hydrothermal alteration of volcano-sedimentary clay-rich sequences. The Jadar deposit has the potential to supply more than 10% of the current global demand for Li. Other rocks, such as some manganese and bauxite deposits, may also host appreciable amounts of this metal, but none is currently considered economic.

Italy is well known for the lithium minerals occurring in the Late Miocene Lithium–Cesium–Tantalum (LCT) pegmatites from the classic locality of Elba Island, Tuscany (Figure 1). Elbaite (Elba is the type locality) and petalite specimens from these pegmatites are represented in mineralogical museums worldwide [12], and, shortly after its discovery, lithium was also separated by Italian chemists from Elba lepidolite [13].

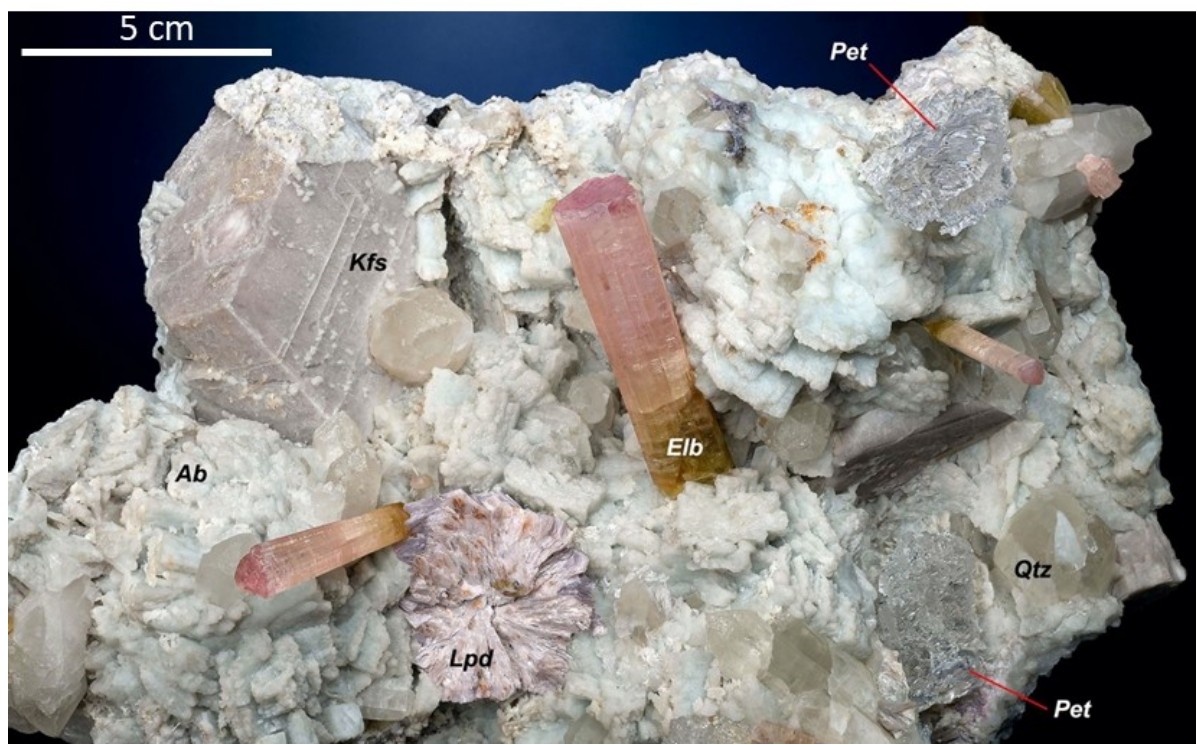

**Figure 1.** A druse from the San Piero LCT pegmatites (Elba Island, Tuscany) showing their typical association of lithium minerals: elbaite (Elb) pink crystals; pale violet aggregates of lepidolite (Lpd); and colorless, partially corroded petalite (Pet) crystals. Kfs: orthoclase; Ab: albite; Qtz: quartz. The biggest elbaite crystal is 6 cm long. Marco Lorenzoni collection and photo (reproduced with permission from the author).

Despite the presence of this famous locality, Italy has no record of lithium production. Nonetheless, the current geopolitical context suggests the opportunity for a systematic appraisal of resources, including innovative, unconventional ones. It is worth mentioning

that Italy recently became a producer of Li-ion batteries [14]; another very large plant is expected in 2024 [15], and a third one may appear soon [16]. The assessment of lithium potential in Italy must account for the conceptual models developed on lithium deposits worldwide, evaluating how they fit the Italian geological and geodynamic setting. In this communication, we review the known lithium occurrences in Italy, with an emphasis on its potential recovery from geothermal fluids. Indeed, a research permit for lithium was recently granted for one of the localities described later [17].

## 2. Geological Background

Italian geography/geology is dominated by two different mountain chains: the Alps, bounding the country to the north, and the Apennines, which outline its southward extension into the Mediterranean Sea (Figure 2). The Apennines continue through Sicily into the Maghreb mountain chain. The Tyrrhenian marine basin, a submerged prominent characteristic of the Italian region, is placed in the middle of the study area, between the arcuate chain of the Apennines to the east and the island of Sardinia to the west. The geological record of Italy reflects the geodynamic evolution of the Mediterranean region during the progressive approach of two main tectonic plates: Europe, to the north, and Africa, plus the Adria microplate, to the south [18]. The Alps and Apennines were formed sequentially through the closure of the Mediterranean Mesozoic Tethyan basins along two opposite subduction zones [18]. The Alps are related to the eastward–southeastward subduction of Europe underneath the Adria microplate (Cretaceous to Present), whereas the Apennines were generated by the westward subduction of Adria underneath Europe (Eocene to Present). During the past 40 Myr, the eastward migration of the Apennine belt and the roll-back of the subducting Adria plate led to the opening of the Provencal and Tyrrhenian back-arc basins. The opening of the Tyrrhenian basin (since 15 Ma) produced progressive thinning of the Paleo-Apennine continental crust and, locally, the formation of a new oceanic lithosphere (Southern Tyrrhenian Sea). During such a fast extensional process, the European Plate was left to the west, with the island of Sardinia and part of the Calabrian arc representing disrupted blocks of the European Variscan basement.

The Alps are morphologically more elevated (average 1500 m, up to 4800 m) than the Apennines (about 500 m, up to 2900 m). Moreover, the Alps were affected by much larger erosion, because uplift in the Alps was much longer and exhumation more efficient than in the Apennines. The Paleozoic basement is much more involved in the Alps than in the Apennines, where the tectonic stack of units is mostly made by the Mesozoic–Tertiary sedimentary covers of the Adria Plate and the Tethyan oceanic domain. The Mesozoic sedimentary sequences of the Tethyan oceanic domains and the Adria continental passive margin were deeply involved in the tectonic structure of both chains, but in the Apennines, they experienced a lower degree of regional metamorphism.

The Paleozoic basement of the European Plate crops out extensively in Sardinia, in the external tectonic units of Western Alps, and as an overthrust unit in the Southern Apennines (Calabrian arc). The Paleozoic rocks were involved in the Variscan orogeny, for which variable metamorphic effects were recorded that range from very high-grade (migmatites) to low-grade (greenschist).

Post-Variscan magmatism is widespread and dominated by Carboniferous calc-alkaline plutons and batholiths, while peraluminous, crustal-derived granite intrusions are less common, and usually emplaced during the Permian.

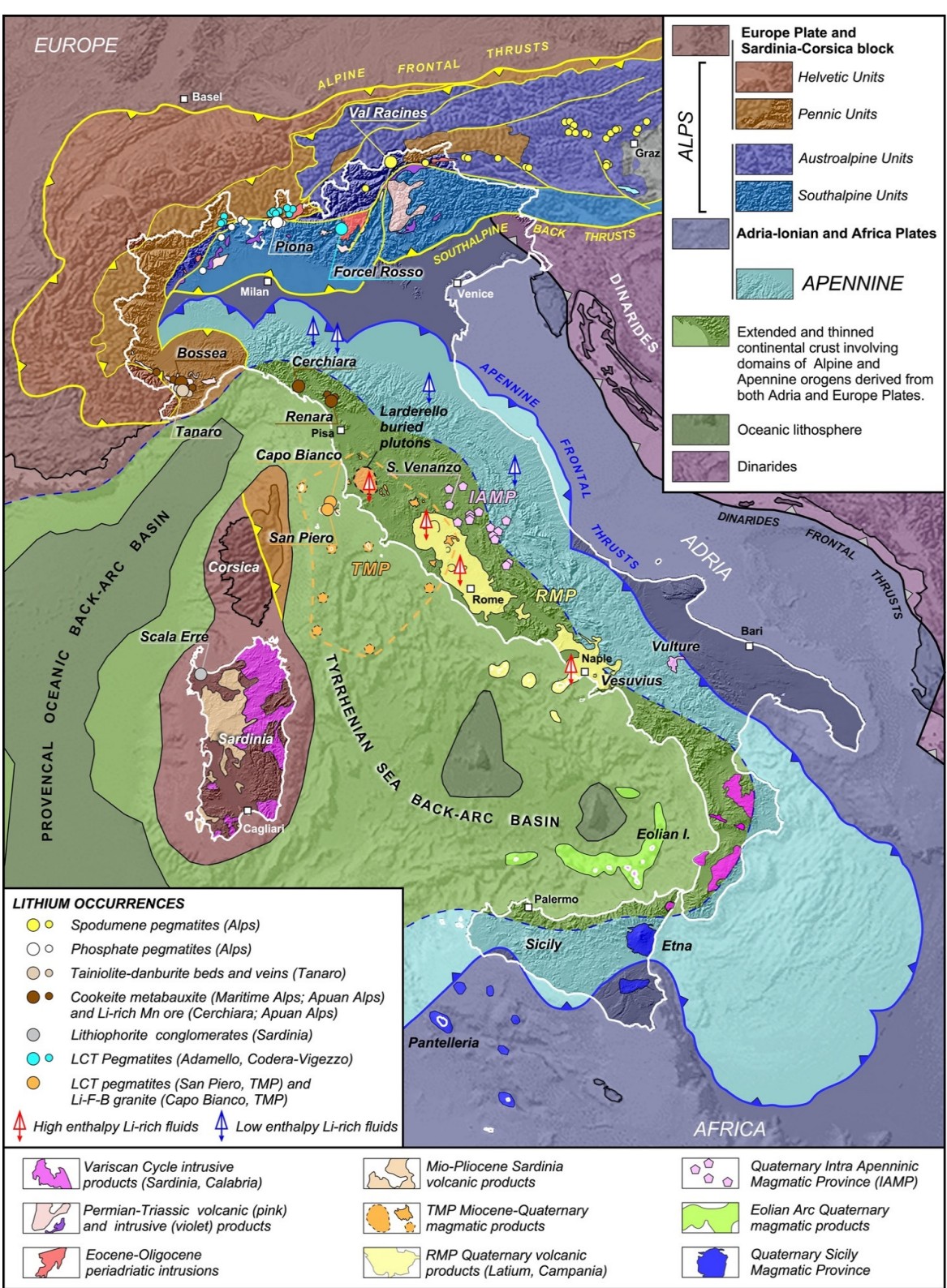

**Figure 2.** Synthetic structural map of Italy showing the hard-rock lithium occurrences and the main magmatic provinces. Modified after the Structural Model of Italy [19]. Details in the text.

The Paleozoic basement of the Adria Plate is well exposed in the Southern Alps (along the Insubric–Periadriatic tectonic lineament), while in the Apennines, it crops out only in small tectonic windows (Apuan Alps, Elba Island, central–Southern Tuscany). Deep exploratory wells (for geothermal resources and oil and gas) intercepted the Adria

basement over a large area extending from the Adriatic to the Tyrrhenian Seas [20]. In the exposed portions of this basement, the metamorphic grade of the Variscan event increases from Central Italy towards North–Northwest, reaching the maximum grade in the Ivrea-Verbano zone (granulitic facies). The presence of post-Variscan magmatism is less evident than in the European plate. Specifically, Carboniferous calc-alkaline activity is lacking, while Permian–Triassic magmatism produced significant I-type granite intrusions in the Southern Alps; some boron-rich, peraluminous subvolcanic intrusions in the Apuan Alps basement; a swarm of spodumene pegmatites in the Austroalpine units [21]; and a large felsic porphyritic volcanic province in the Trentino Alto Adige [22]. The overall magmatic and metamorphic characteristics are consistent with the peripheral position that most of the Adria basement had during the Variscan and post-Variscan metamorphic–magmatic events.

The differences between the Alps and the Apennines also concern Tertiary–Quaternary magmatism. Subduction-related volcanism in the Alps was very limited, while along the Apennines, there was significant volcanic activity before (Sardinian Oligo–Miocene cycle; Figure 2) and during/after (Central–Southern Italy, Pliocene–Quaternary cycle) the opening of the Tyrrhenian back-arc basin [18]. Several Eocene–Oligocene calc-alkaline plutons occur in the Alps (Figure 2), but few are crustal-derived intrusions, whereas the back-arc region of the Central–Northern Apennines is characterized by a widespread peraluminous, crustal-derived magmatism and few calc-alkaline/shoshonite/lamproite products (Tuscan Magmatic Province, hereafter TMP; [23]). Finally, the Quaternary magmatism that characterizes the volcanic areas of Central–Southern Italy (Figure 2) is dominated by mantle-derived igneous products of highly variable affinity [18,24–26]: from calc-alkaline/shoshonitic to ultrapotassic and sodic alkaline.

In Europe, Li deposits occur throughout the geological time and tectonic domains, but appear concentrated in specific temporal and spatial segments [10]. Except for the peculiar Jadar deposit, the largest resources (>60% of the estimated tonnages) are associated with segments of the Variscan chain (Galicia Tras-os-Montes, French Massif Central, Cornwall, and Bohemian Massif). Other potentially important deposits are associated with the Archean–Proterozoic Ukrainian shield, and minor occurrences are present in the Proterozoic Svecofennian orogeny. Only small occurrences are known in young Phanerozoic Mediterranean orogeneses.

Given this context, it is presumed that the potential of Italy for significant Li resources is low, although it was never systematically explored. The essential geological, geo-morphological and climatic characteristics for the formation of salars are obviously lacking, and the restricted exposure of the pre-Mesozoic basement limits the possibility of finding conventional "hard rock" deposits (spodumene pegmatites). However, geochemical evidence points to the occurrence of several distinct anomalies in stream sediments (Central Italy, Northern Sardinia, Calabria, and Sicily) and floodplain deposits (Northwestern Italy and Southeastern Sardinia) [27,28]. The application of the cell-based association (CBA) method [29,30] suggests a highly favorable context in Calabria. Moreover, unconventional lithium resources, such as geothermal fluids and low-grade magmatic rocks containing leachable and/or separable Li-rich micas could represent a significant target for scientific research and mining exploration in Italy in the near future. The widespread thermal anomalies related to the recent magmatism all along the western side of the Central–Southern Apennines mark the area of maximum potential. In particular, the peculiar geochemical characteristics (see Section 3.1) of volcanic rocks from the Roman Magmatic Province (Latium and Campania) could be pivotal in determining high-priority exploration targets. The following section contains descriptions of Italian Li occurrences; they are listed in Tables S1 and S2.

## 3. Main Occurrences

### 3.1. Magmatic-Related Occurrences

The oldest (not considering the small Infracambrian orthogneiss bodies in Calabria [31]) magmatic cycle documented in Italy is the Upper Cambrian to Middle Ordovician bimodal

magmatism, which was widespread in pre-Variscan Europe, e.g., [32]. Metamorphosed rocks of this cycle occur throughout the Alps, in Sardinia, Calabria, Tuscany, and Sicily. In Sardinia, this magmatism is well expressed through at least three cycles [33], and is a testament to the development of long-lasting volcanic arcs. At the speculation level, felsic volcanism and its associated continental sedimentary basins may have the potential for Li resources. However, in Europe, Li deposits of this age are reported only in the Avalonian domain of Ireland and Scotland [10].

Late Paleozoic magmatic rocks of the Variscan cycle are, in principle, more attractive because, as said previously, they host significant Li resources at the European scale. Intrusive rocks of this cycle extensively occur in the Sardinia–Corsica block, in the Alpine nappes, and in the Calabrian—Peloritan arc. Along with the Maures–Estérel massif, these are portions of the so-called "Southern Variscides". The Southern Variscides have been reworked by rigid block rotations and translations since the Permian era, and locally involved in the tectono-metamorphic processes of the Alpine orogeny [34,35]. The correlation of the "Southern Variscides" to the litho-tectonic domains established in Western Europe is still disputed [35], in spite of some general affinities with the French–Iberian domains [36–38].

The oldest Variscan magmatic event (345–337 Ma; high Mg-K granites) is recorded in Northwestern Corsica only, while the two subsequent, volumetrically larger events (322–300 Ma and 295–270 Ma; [38]) are also well represented in the Sardinia basement. Sardinia granitoids vary in composition from gabbro to monzogranite, with minor leucogranite bodies. Their chemical affinity ranges from metaluminous to slightly peraluminous, but some large (100 km$^2$), strongly peraluminous intrusions have been described [39]. Felsic intrusions increased with time, and they acquired a widespread F-bearing characteristic in Southern Sardinia [40,41]. Geochemical and isotopic data indicate that Late Paleozoic Sardinia granitoids were mostly derived from magmas generated in the lower continental crust via the partial melting of metasedimentary and metaigneous protoliths [38,39,41], and were eventually contaminated by a small component of mantle-derived melts [42].

There are very few data on the Li contents of Sardinian Variscan rocks. Lithium is not routinely determined in whole-rock analyses, which are often affected by X-ray fluorescence and/or following fusion with Li metaborate. Moreover, while there is a quite extensive literature on the Variscan plutonic rocks of Sardinia [38,43] and references therein, systematic studies of pegmatites are scarce. The most comprehensive study remains that of [44]. According to the author, the Li contents of pegmatitic muscovite range from 100 to 1000 µg/g; pegmatites in the Nuoro area (N. Piscapu and Buddusò) contain Li minerals (elbaite, triphylite, ferrisicklerite). A peculiar type of fayalite-bearing pegmatite occurs at Quirra; it contains up to 151 µg/g Li (whole rock; [45]). Data for leucogranites and pegmatites at Porto Ottiolu, NE Sardinia are reported by [46]. The Li contents range from 8 to 56 µg/g (presumably; the table listing the analytical data does not specify units). Interestingly, with respect to leucogranites, pegmatites are enriched with Ba, Rb, and Cs, but have lower Li contents (8–40 µg/g compared to 34–56 µg/g). Additional information on Li minerals in Sardinia includes: (i) tourmaline in the Mandrolisai magmatic complex has up to 0.239 atoms per formula unit (apfu) of Li (3600 µg/g; [47]); (ii) by contrast, tourmalines associated with the Arbus pluton have very low Li (700 µg/g max; [48]); the authors conclude that the tourmaline evolution trend (schorl $\geq$ foitite) is consistent with Li-poor granitic melts; (iii) biotite from F-bearing granites of Monte Linas (Southern Sardinia) has up to 1.95 wt% Li$_2$O (1.14 apfu; [40]). Finally, there is an unconfirmed report of "zinnwaldite" in a greisen assemblage at Gonnosfanadiga [49]. In summary, there is only sparse, non-systematic information on the occurrence of Li in Sardinia. A re-assessment of the potential for this metal, especially in the peraluminous complexes, would be desirable.

According to [50], Late Variscan magmatism in the Calabrian–Peloritan arc is occurred in a time span between 320 and 280 Ma. Specifically, a trondhjemite intrusion occurred at ~314 Ma, followed by peraluminous leucogranites (304–300 Ma), and finally, by the composite (gabbro to leucogranite) Serre and Sila batholiths (297–292 Ma; [51–53] and references therein); the late pegmatite dykes may be as young as 265 Ma [54]. The whole magmatic

suite has been interpreted as the result of the recycling and reworking of different crustal sources [51,52]. As previously noted, [30] identified Calabria (notably, the metamorphic and intrusive lithologies) as highly favorable for lithium mineralization. Indeed, a stream sediment in northeastern Sicily returned a Li value of 92 µg/g, one of the highest reported for Italy [27,28]. However, actual data on Li occurrence in rocks are even scarcer than for Sardinia. Detailed pegmatite descriptions are also lacking; specifically, we could not find any explicit mention of Li-bearing minerals. The authors of [55] included Li in a few multi-element analyses of sulfides from the mineral deposits of the Peloritani mountains. Not surprisingly, the reported contents are very low (max 14 µg/g).

Magmatic rocks of the Late Cambrian—Ordovician and Variscan cycles are also well represented in various tectonic units of the Alps, e.g., [56], but the only documented Li resources are associated with late- to post-orogenic Permo–Triassic magmatism. In the Austroalpine and Southalpine units of the Alps, the majority of the isotopic ages available for the largest volumes of Permian–Triassic magmatic rocks fall in the range 290–260 Ma, more than 15–30 My younger than the latest phases of Variscan deformation, e.g., [57,58]. Such a temporal gap indicates that the formation of Permian basins and the associated magmatic products could have been caused by extension during post-orogenic crustal relaxation, or by large-scale strike-slip movements along the southern margin of the Variscan orogen. The large volcanic–subvolcanic (Sesia, Lugano–Valganna, Athesian) and intrusive (Laghi granites, Torgola–Navazze, Brixen–Iffinger–Mt. Croce, Cima d'Asta) complexes are, apparently, not associated with either lithium ores or significant occurrences of lithium minerals. Zinnwaldite has been reported from miarolitic cavities of granophyric subvolcanic bodies (Cuasso al Monte) and pegmatites from the Baveno–Mottarone granite intrusions [59]. However, lithium concentration in felsic rocks was never investigated systematically. Furthermore, the strict association of rhyolites and continental sediments in the Early Permian basins (such as the Collio formation in the Brescian and Bergamasc Alps and similar formation in the Eastern and Western Alps) represents an optimal environment for the remobilization of lithium from volcanites and concentration in continental sedimentary evaporitic basins (see next section on sediment-hosted occurrences).

The Paleozoic basement outcrop in Tuscany is mostly represented by metasedimentary formations and metaigneous formations of the Late Cambrian–Ordovician cycle, which experienced low-grade metamorphism during both the Variscan and Alpine tectono-metamorphic events [60]. The Ordovician, metaigneous felsic rocks of the Apuan Alps have remarkable similarities with those of Southern Sardinia and, likewise, were never investigated for their lithium potential.

The most relevant occurrence of hard-rock lithium ores in the Alps is represented by the Upper Permian–Early Triassic spodumene pegmatites (250–270 Ma) hosted by the Austroalpine units (Figure 2; [21]). They are part of a 400 km-wide pegmatite field, from Graz (Austria) to Mt. Cevedale (Italy), hosting thousands of relatively small pegmatite and leucogranite dykes, generated by the partial melting of metasedimentary crust during the so-called "Permian metamorphic event" [61]. This event is a consequence of lithospheric thinning accompanied by magmatic underplating of the lower crust, during the extensional processes that affected the Variscan orogen before the opening of the Early Triassic Meliata Ocean.

Most of the spodumene pegmatites are uneconomic, but in the Wolfsberg District (Carinthia, Austria; [62,63]) an exploration permit has been active since 2019, and a pre-feasibility study indicated resources of 6.3 Mt at 1.17% $Li_2O$ (the start of production is scheduled for 2023; [64]). A few spodumene pegmatites have also been described in Alto Adige, Northern Italy, close to the Austrian border (Figure 3; [21,65,66]): Val Martello (Meran), Val Racines (Vipiteno), and Uttenheim (Val Pusteria).

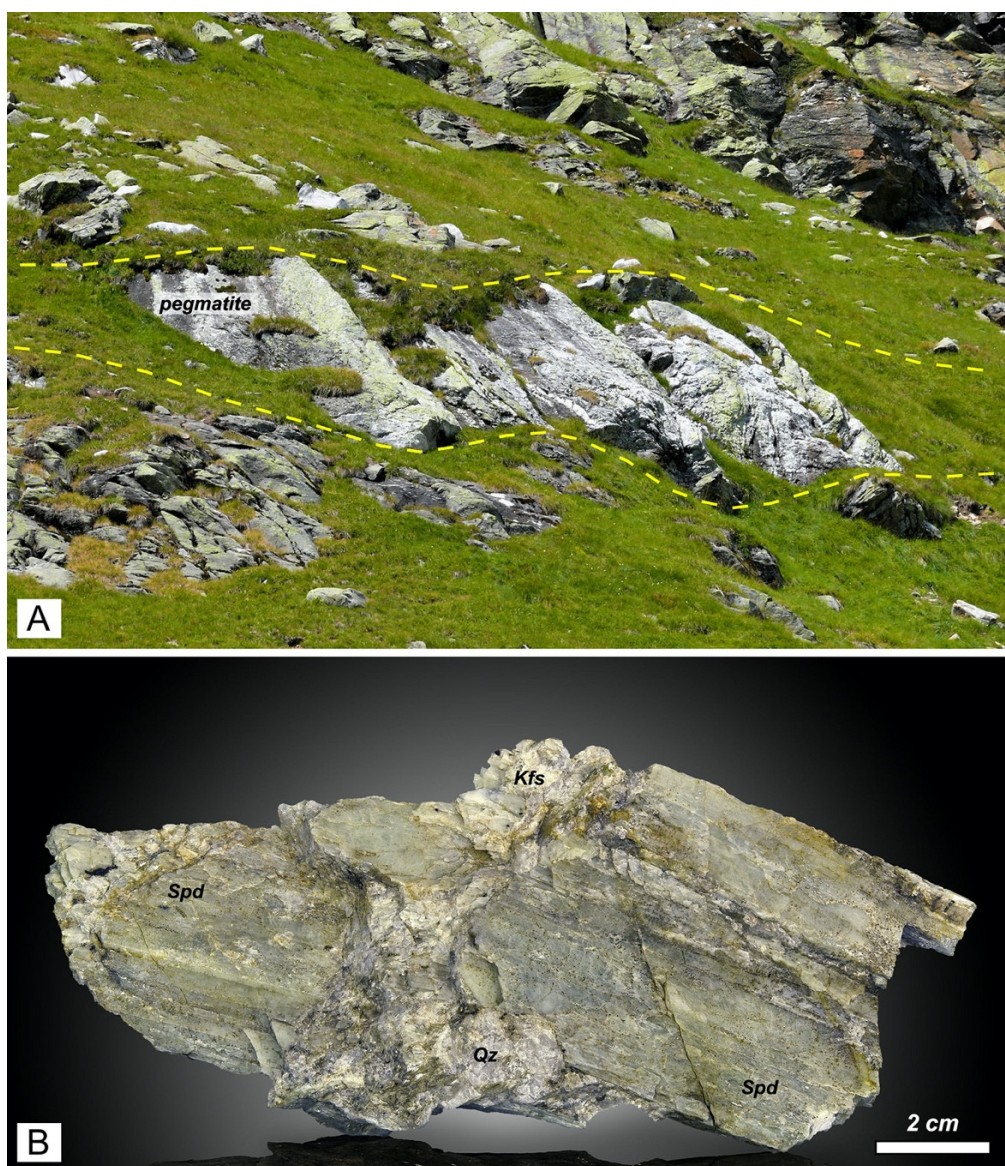

**Figure 3.** (**A**) Outcrop of a spodumene pegmatite body at Hohe Kreuzspitze, Val Racines (BZ). Photo by Mirko Grisotto (reproduced with permission from the author); (**B**) spodumene (spd) prismatic crystals up to 16 cm long, embedded in quartz (Qz) and feldspar Kfs), from the Hohe Kreuzspitze pegmatites. Photo and collection from Mirko and Lodovico Grisotto (reproduced with permission from the authors).

Further to the west, from Mt. Cevedale to the Sesia-Ossola valleys, there is a belt of about 200 km where the basement of Southalpine units hosts numerous small felsic dykes, ranging in age from Upper Permian to Early Jurassic (257–190 Ma; [57,67–70]). Phosphate-rich pegmatites occur at Malga Garbella, near Mt. Cevedale (257 Ma; [69,71]), as well as at Brissago, north of the Italian–Swiss border (242 Ma; [68]), and near Piona, on Como Lake (229–208 Ma; [67,72–74]). Most of these granite pegmatites are known for the occurrence of rare minerals, and some of them have been exploited for raw ceramic materials. They are characterized by complex parageneses (schorl, garnet, beryl, muscovite, chrysoberyl, zircon, uraninite, monazite, columbite, tapiolite), involving a number of rare Fe-Mn-Ca-Na-Al phosphates. Li occurrence is restricted to small amounts of late phosphates (triphylite, lithiophilite, ferrisicklerite; [73]); the potential as a lithium resource is, therefore, low.

Evidence of a Middle Permian magmatism (tourmaline-bearing felsic metaigneous rocks) was recently found in the basement of the Apuan Alps (Northern Tuscany; [75,76]).

The Permian magmatism mostly affected the Variscan basement, and possibly Permian sediments, producing widespread tourmalinization and phyllic alteration, with local formation of small Pb-Zn-Ag and barite–pyrite orebodies [76,77]. Geochemical analyses and mineralogical studies [78,79] highlighted the Sn-W-Be anomalous character of the ores, but lithium potential was not addressed.

Such a Permian magmatic–hydrothermal system shows some analogies with Permian granite–ore systems of the Southalpine units, e.g., Torgola-Navazze [80]. These Sn-W-bearing systems may host zinnwaldite-bearing greisens.

The Tertiary–Quaternary magmatism of Italy is one of the most varied, studied, and debated petrological–geodynamic subjects on the planet. In this section, we will only focus on those magmatic complexes which, due to their mineralogical and/or geochemical characteristics, can be considered of interest for their lithium potential. The available data in the scientific literature indicate that lithium minerals and/or anomalous lithium concentration in rocks occur in the following Tertiary–Quaternary magmatic complexes: (1) pegmatites in the contact aureole of the Adamello intrusion (Central Alps; Periadriatic Magmatic Province); (2) pegmatites related to the Lepontine thermal dome (Central Alps; Periadriatic Magmatic Province); (3) granites and pegmatites of the Tuscan Magmatic Province; (4) volcanic rocks of the Roman and intra-Apenninic magmatic Provinces.

The Periadriatic magmatic rocks were emplaced in the Eocene–Oligocene era along the Insubric Fault, mostly exploiting structural traps in the Southalpine, Austroalpine and Penninic units ([25,81] and references therein). From west to east, the main intrusive complexes are: (1) the Traversella pluton (monzodiorite-gabbro; ca. 31 Ma); (2) the Biella-Valle del Cervo pluton (monzonite-syenite; 31–30 Ma); (3) the Miagliano pluton (diorite-tonalite; ca. 33–30 Ma); (4) the Bregaglia (Bergell) pluton (granodiorite-tonalite; ca 32–30 Ma) and the satellite Triangia pluton; (5) the spatially associated, but younger (24 Ma), Novate-San Fedelino leucogranite; (6) the Adamello batholith (tonalite-diorite-granodiorite; 43–31 Ma); (7) the Rensen-Monte Alto pluton (tonalite; 32–31 Ma); (8) the Cima di Vila pluton (tonalite; 32–31 Ma); and (9) the Vedrette di Ries pluton (tonalite; 32 Ma). Apart from the Adamello batholith and the Bregaglia pluton, the other intrusive bodies are relatively small in size, and their emplacement took place over a limited period of time. All these intrusions display a calc-alkaline geochemical and isotopic signature, with minor involvement of crustal components; the contact aureoles are highly variable in size and metamorphic grade.

Plutonic rocks and their contact aureole commonly host pegmatite dykes, but lithium-rich minerals have only been observed in a single locality at the south-western border of the Adamello batholith (Forcel Rosso, Valle Adamé; [82,83]). The sub-horizontal granite pegmatite dykes, tens of meters in length and 1–2 m thick, crosscut the Permian–Triassic sequence (Southalpine units) at few hundred meters from the contact with the Adamello tonalite [84]. They display strong asymmetric zoning with a medium-grained, layered bottom unit and a coarse-grained upper unit containing schorl, garnet, and muscovite. Miarolitic cavities locally occur in the axial zone, where lithium-rich minerals are concentrated (polychrome elbaite-liddicoatite, pinkish lepidolite; [82]).

A larger group of Alpine pegmatites crop out in the Central Alps, along the so-called "Lepontine metamorphic dome", a zone to the north of the Insubric line, where the highest metamorphic grade (Barrovian to migmatitic) was reached during the Alpine event [85,86]. These pegmatites span an E–W trending belt, about 100 km in length, from the Bregaglia pluton (east) to the Ossola Valley (west). They are hosted by the Penninic units as well as by the intrusive rocks of the Bregaglia pluton. Pegmatite dykes are especially abundant in the so-called Southern Steep Belt, close to the Insubric line, becoming rarer northward. Isotopic ages indicate a prolonged emplacement process (29–20 Ma), overlapping the formation of the nearby peraluminous Novate granite pluton (24 Ma), and cutting through the Bregaglia pluton (32–30 Ma) and the high-grade fabric of the metamorphic host rocks. Most pegmatite dykes have a simple mineral assemblage consisting of K-feldspar, quartz, muscovite ± biotite, and scarce albite. A few dikes contain rare accessories. including almandine–spessartine garnet, blue beryl, schorl, and, locally, phosphates and Nb-Ta-Sn-Y-

REE-U oxides. Lithium minerals are rarely observed, and include Mn-rich elbaite [85] and ferrisicklerite (at Val Bodengo; [87]); lepidolite is also reported [49].

The typical products of the Late Miocene–Quaternary Tuscan Magmatic Province (TMP) are plutonic, subvolcanic, and volcanic felsic rocks, mostly derived from partial melting of the Adria metasedimentary Paleozoic basement, while only a minor contribution came from mantle-derived magmas ([88] and references therein). Magmatism was diachronous, starting in the Late Miocene in the western part of the province and progressively migrating eastward, where it is presently active below the Larderello geothermal field [89]. TMP hosts many Li- and B-rich peraluminous granite intrusions [23,90,91], but significant LCT pegmatites have been observed only at Elba Island (San Piero; Figures 1 and 2). As previously noted, the locality has long been known for its beautiful specimens of polychrome elbaite, pollucite, pink and blue beryl, and spessartine, which have been extracted since the end of the 18th century [12]. Tens of small pegmatite dykes (10–20 m long; up to 1 m thick) crop out along the eastern border of the Mt. Capanne monzogranite pluton, also propagating into the contact aureole [91]. In addition to elbaite, other lithium minerals are represented by lepidolite, petalite, rossmanite, tsilaisite, and amblygonite. Lithium minerals are concentrated in the pocket zone at the core of the pegmatite dyke.

However, the overall Li-rich characteristic of the pegmatitic magma is also confirmed by lithium concentrations of up to 4000 μg/g in siderophyllite, crystallized earlier in the border zones [92]. Lithium in TMP felsic rocks (Figure 4, Table S1; 40–380 μg/g; [23,90,91,93–95]) is mostly hosted by aluminiferous biotite (up to 3000 μg/g) and, in few cases, by white micas (up to 2200 μg/g) and schorl-dravitic tourmaline (up to 2500 μg/g).

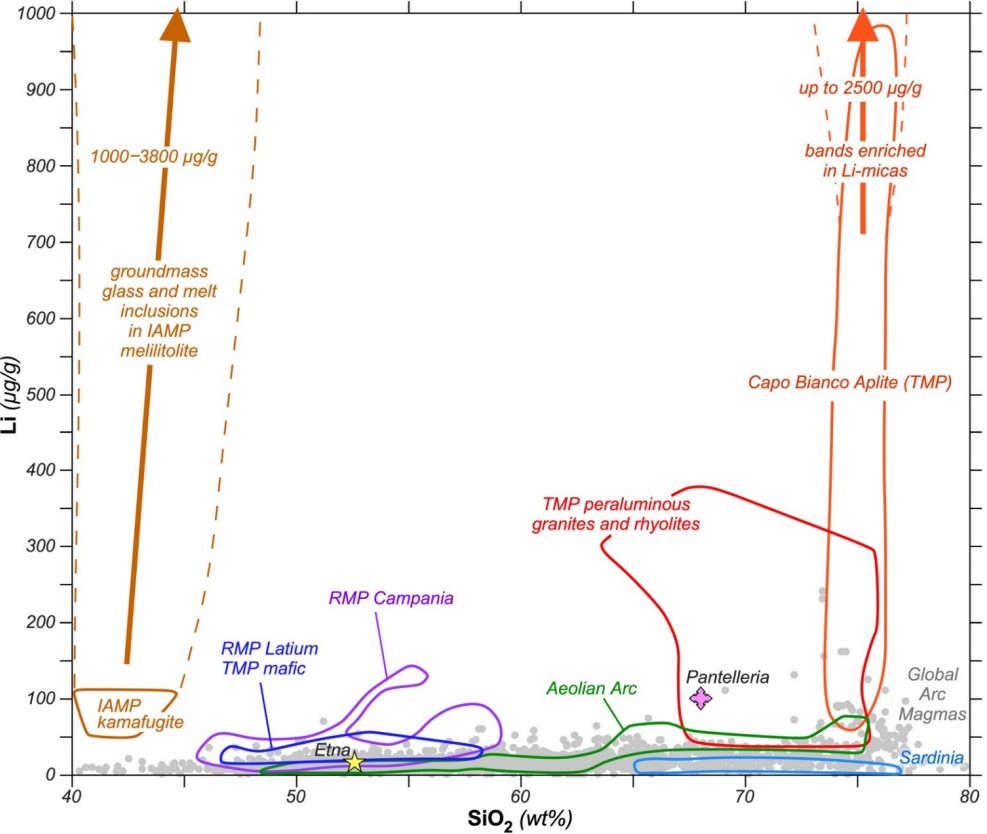

**Figure 4.** Li vs. silica content of magmatic rocks (whole rocks were not specified) from Central—Southern Italy compared with global arc magmas. Sources of data for Italian rocks: RMP Latium and TMP mafic rocks [96,97]; RMP Campania [98,99]; TMP granites, rhyolites, and Capo Bianco Aplite [[23,91–93,95] and this work; intra-Apenninic Magmatic Province (IAMP) [96,100]; Pantelleria [101]; Sardinia [102]; Aeolian Arc [103,104]; and Etna [105,106]. Source of data for global arc magmas: compilation by [107].

The TMP felsic rock with the highest lithium content (up to 1000 μg/g in the whole rock; Table S1) is the Capo Bianco Aplite at Elba Island (Figure 5; [91,92,95]). This strongly peraluminous alkali feldspar granite forms a few sills of ca. 0.5 km³ that were emplaced at a shallow crustal level during the Late Miocene [108,109]. The Capo Bianco Aplite is characterized by phenocrysts of Li-rich muscovite, Li-rich siderophyllite, K-feldspar, oligoclase, and quartz, embedded in a cryptocrystalline groundmass made of albite, k-feldspar, and quartz [92].

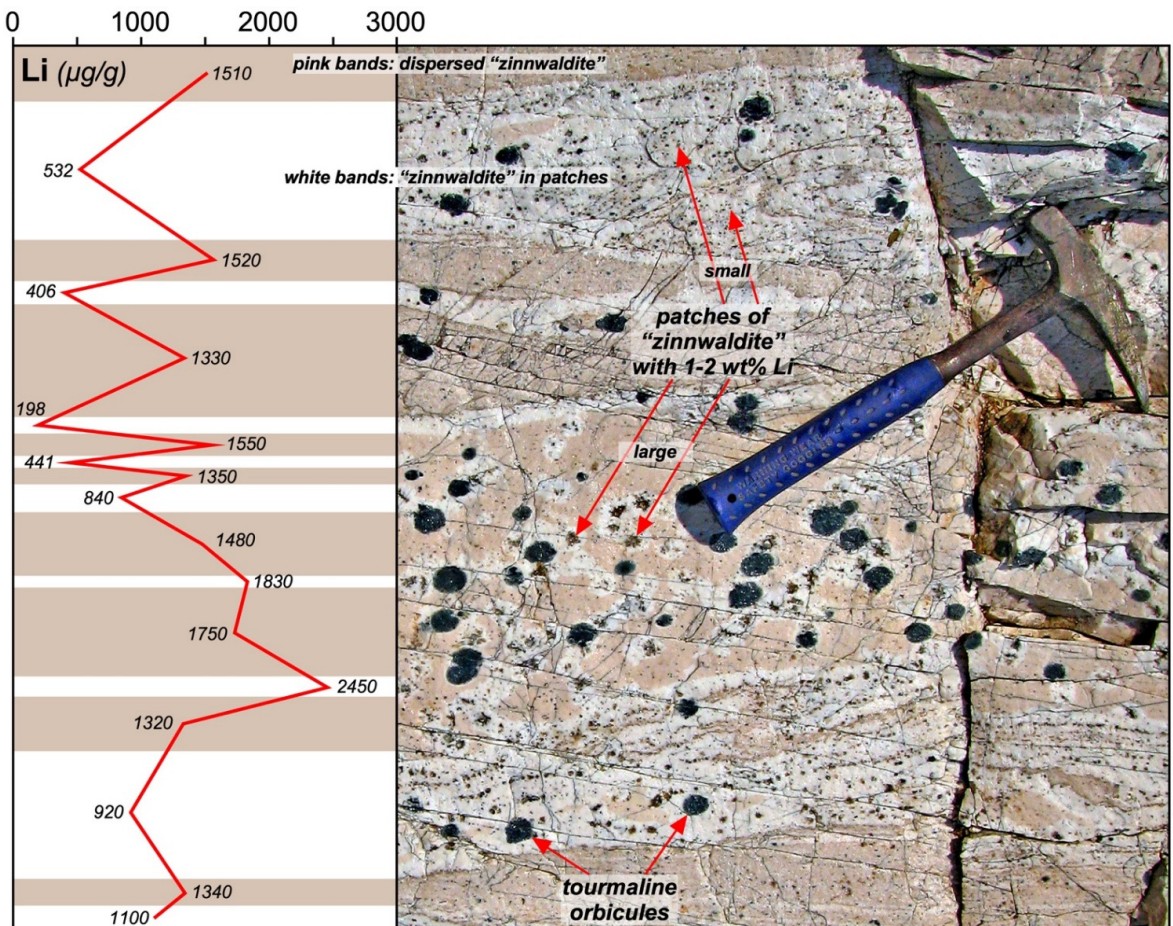

**Figure 5.** The magmatic layering of the Capo Bianco Aplite (Elba Island, Tuscany) is highlighted by the variable content and texture of Li micas, leading to the characteristic pink and white banding. The analyses of lithium in each band (Actlabs Ltd. analyses via four-acid digestion and ICP-MS; this study) are shown in the graphic to the left. Tourmaline orbicules occur in both pink and white bands; the composition is schorlitic, with local evolution towards elbaitic composition.

Large portions of the tabular intrusions show marked magmatic layering, with rheomorphic structures that are outlined by the alternation of bands with variable content of Li-micas ("zinnwaldite") and tourmaline orbicules (schorl–elbaite). Individual centimetric to decametric thick bands, strongly enriched in Li-micas, may reach lithium contents up to 2500 μg/g (Figures 4 and 5; data from this study). High-temperature, Li-rich fluids were also identified in fluid inclusions from hydrothermal veins cored on top of buried TMP plutons in the Larderello geothermal field [110,111], and in water-rich melt inclusions from Elba Island LCT pegmatites [112].

The last group of Italian magmatic rocks showing an anomalous lithium content is represented by the potassic and ultrapotassic, Quaternary volcanic rocks of the Roman (RMP) and intra-Apenninic magmatic provinces (IAMP). They range in composition from kamafugite (IAMP) to basanite–trachybasalt and phonolite–trachyte (RMP). A few studies

investigated the lithium content of these rocks. The first studies that addressed the lithium geochemistry of mantle-derived rocks from TMP, RMP, and IAMP were [96,97,99]. They found Li concentration up to 90 µg/g and 105 µg/g, respectively, in RMP and in IAMP whole rocks (Table S1). Similar, and even higher concentrations were found in RMP rocks from the Vesuvius–Phlegrean area (up to 140 µg/g; [98]). Comparing these data with the composition of Global Arc Magmas [107], it is evident that, for equal silica content, volcanic rocks from RMP and IAMP are systematically enriched with lithium (Figure 4).

Such anomalous behavior is also obvious when these rocks are compared with volcanic rocks from other Italian localities (e.g., Aeolian Arc, Sardinia; [102–104]. The reason(s) for this difference should be investigated. The behavior of lithium during and shortly after an eruption is relatively unconstrained [113], but its speciation, especially in pyroclastic rocks, seems to be controlled by three main processes [114]: (1) a pre-eruptive lithium pile-up in the melt driven by fractional crystallization; (2) the syn-eruptive degassing/exsolution of lithium-rich magmatic fluids; and (3) the post-eruptive condensation of lithium from magmatic fluids onto dust particles or glass shard surfaces. In metaluminous magmas, the incompatibility of Li in the structure of nearly all magmatic minerals should result in relative enrichment in the glass component, e.g., [100,101]. Such behavior seems to be confirmed by the extremely high lithium concentration found in [100] in the interstitial glass, and in melt inclusions studied in the kamafugite–melilitolite from San Venanzo (IAMP). The preferential incorporation of lithium in easily leachable volcanic glass, or its condensation on the particle surfaces of pyroclastic sequences, could represent a relevant characteristic to be considered for the lithium potential of geothermal brines/fluids in central Italy, as discussed in Section 3.3. There is an obvious need for additional studies on lithium geochemistry and speciation in volcanic rocks from central Italy, as well as in the sedimentary-metamorphic sequences hosting the volcanic plumbing systems and the geothermal systems.

### 3.2. Sediment-Hosted Occurrences

#### 3.2.1. Mn Deposits

The chert-associated Mn ores of Eastern Liguria, hosted within siliceous sediments, are the largest Mn producer in Italy [115]. The most important district is located in Val Graveglia (e.g., the Gambatesa, Molinello, and Cassagna mines), where ores are hosted by the Jurassic oceanic radiolarites of the Ligurian units. In spite of the large variety of mineral species, there are no reports of lithium mineral from the Val Graveglia deposits. Conversely, the minor Cerchiara mine, lying east of the main district, is of interest here because it contains several lithium minerals, including newly discovered species (balestraite, [116]; lavinskyite-1M, [117] and aluminosugilite, [118]). The geology of this deposit is poorly known, but the host rocks belong to a continental sedimentary sequence made up of radiolarites, marls, and shales (Falda Toscana; [119]). The deposit is correctly placed in Figure 7 of [10], but erroneously ascribed to Ireland in the Supplementary File. A metamorphic analogue of the Cerchiara Li mineral assemblage occurs in the Apuan Alps, where sugilite has been described in a small Mn occurrence near Vagli [120]. Finally, we mention supergene deposits of lithiophorite occurring as cement and pebble coatings in Messinian alluvial conglomerates at Scala Erre, Northern Sardinia [121]. Lithiophorite was identified via X-ray powder diffraction patterns; bulk Li contents were not reported.

#### 3.2.2. Bauxite

In addition to being our source of aluminum, bauxites were recently the target of many studies to assess their potential for "critical metals", e.g., [122] and references therein. Li is quite a mobile element, and as such, is not likely to be concentrated in bauxites; however, some bauxites have Li contents that represent a potential resource, e.g., [123] and references therein; in Europe, Li minerals are reported in Hungarian bauxites [124]. Italy has non-negligible bauxite deposits in Sardinia, Campania, Abruzzi, and Apulia; some of them were exploited until recently, and were also investigated for their potential for

"critical metals" [122,125]. However, there are no published data on their Li contents. On the other hand, Li-rich metabauxite layers, hosted by metamorphosed Mesozoic carbonatic sequences, are reported to occur in the Briançonnais domain in the Ligurian Alps [126], and from the Tuscan domain in the Apuan Alps, Tuscany (Figure 6; [127]); the Li-bearing phase here is cookeite. The reported whole-rock Li contents are up to 2.3 g/kg.

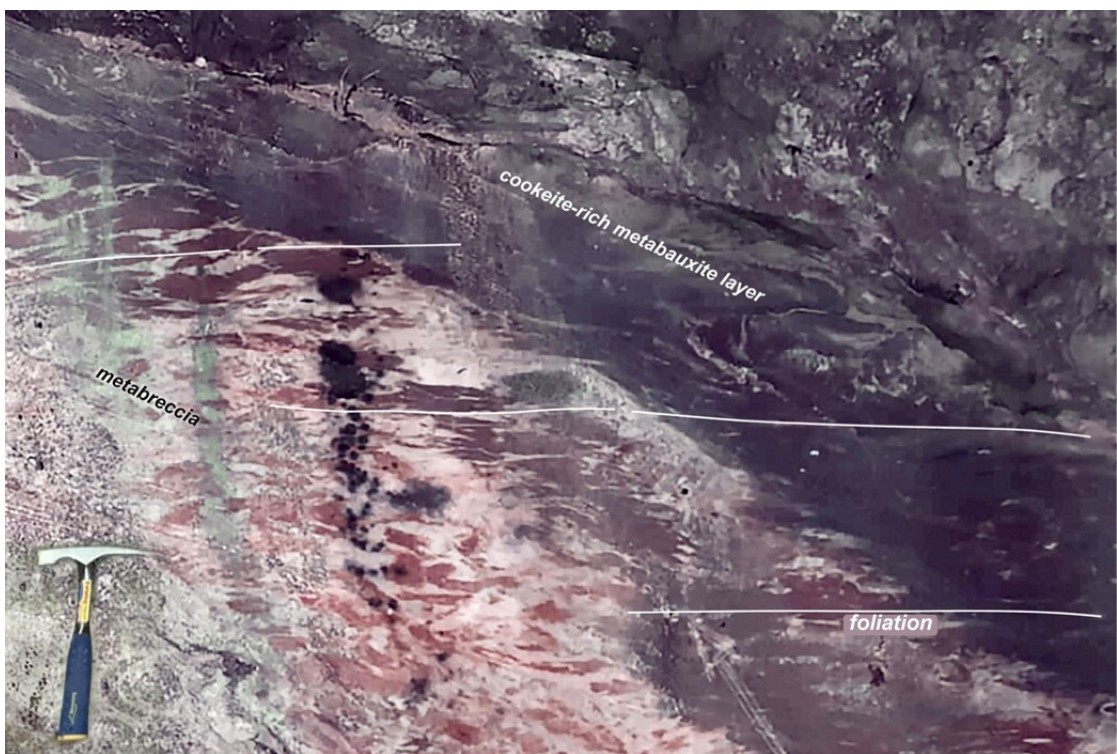

**Figure 6.** The dark red, cookeite-rich metabauxite layer and metabreccia, exposed at the Renara marble quarry, Apuan Alps (Tuscany). They are part of the Mesozoic carbonate sequence (Tuscan domain) metamorphosed during the Apenninic event.

### 3.2.3. Jadar-like Deposits

As noted above, the Jadar deposit occurs in a very peculiar intramontane lacustrine evaporite basin. Lithium enrichment is presumably associated with the leaching of comparatively Li-rich protoliths such as felsic volcanic rocks [10]. In Italy, a potential scenario whereby such a process may have occurred is represented by Lower Permian continental deposits, receiving contributions from nearly coeval volcanic rocks. A typical example is the Collio Formation in Northern Italy [128]. This widespread formation may reach remarkable thickness (>2 km) and is known for the occurrence of uranium and siderite deposits ([129] and references therein). There is no information on Li contents, but we notice the presence of a Li-mica (tainiolite, $KLiMg_2(Si_4O_{10})F_2$), associated with danburite ($CaB_2Si_2O_8$), in the metamorphic equivalent of this formation in Val Tanaro, the Maritime Alps (Figure 7; [130]). The Val Tanaro tainiolite–danburite-bearing rocks are stratigraphically localized in a roughly continuous bed, with a lateral extension of about ten kilometers. The lithium–boron-bearing beds have thicknesses varying between a few decimeters to 5 m, and they are locally associated with danburite–tainiolite–quartz–phlogopite veins [130]. The intimate association of tainiolite and danburite in these greenschist facies rocks could be related to the Alpine metamorphic re-working of a Jadar-like deposit of Permian age. The stability of jadarite ($LiNaSiB_3O_7(OH)$) is poorly known; its breakdown to an assemblage made up of a boron mineral (danburite) and a lithium mineral (tainiolite) is, obviously, highly speculative. The lithium content of this unusual rock is not available; an accurate geochemical-petrogenetic study would be desirable, given the considerable extent of the tainiolite-danburite beds. Unraveling the origin of this rock could reveal new knowledge

on the geochemical cycle of lithium in the volcano-sedimentary, Permian basins and the development of a new conceptual-exploration model in similar settings worldwide.

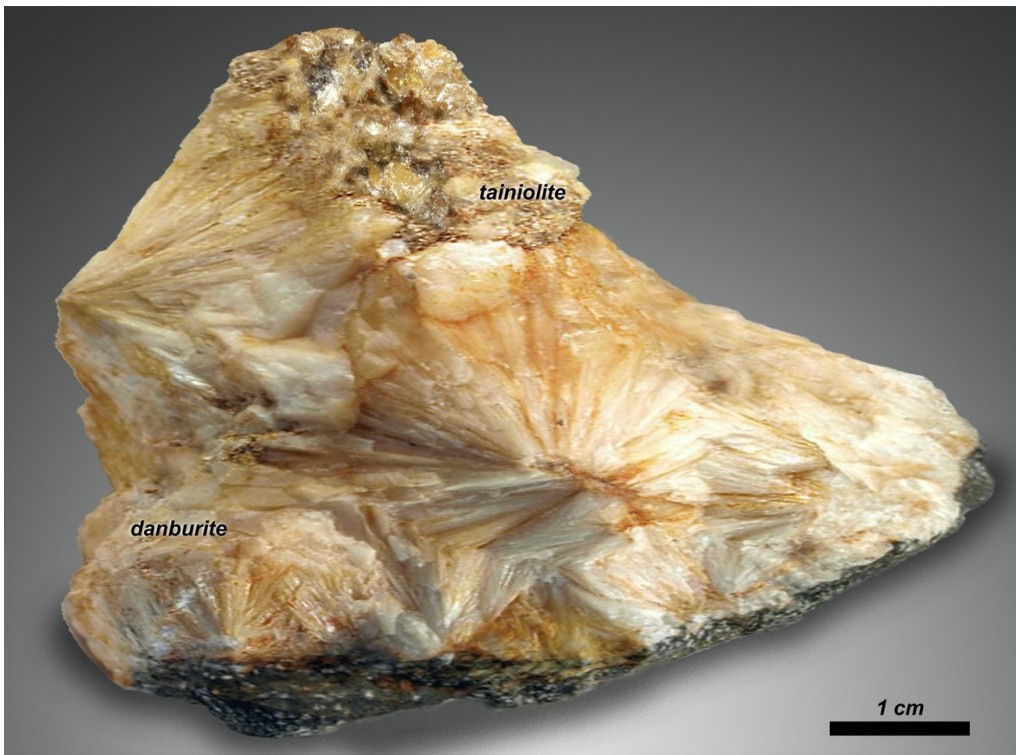

**Figure 7.** Tainiolite, yellowish-brown flakes with fibrous, radiating aggregates of danburite from a vein associated with the Permian lithium–boron-rich beds of Val Tanaro (Piedmont).

*3.3. Continental Waters*

Under this broad title, we include subsurface waters and thermal springs. Specifically, following [131], we will distinguish high- to intermediate-enthalpy fluids (T > 90 °C), and low-enthalpy fluids (T < 90 °C); the latter, in turn, can be subdivided into groundwaters and thermal springs. We emphasize that this distinction serves only descriptive purposes.

3.3.1. High-to-Intermediate Enthalpy

The main high- to intermediate-enthalpy Italian geothermal fields are located in the western parts of Tuscany, Latium, and Campania, in association with Plio–Quaternary magmatic provinces (TMP and RMP; Figures 2 and 8). These areas are characterized by the presence of wide thermal anomalies, with high heat flow (above 100 mW/m$^2$ at the regional scale, with local peaks of up to 1 W/m$^2$ in the geothermal zones). As outlined in Section 2, the thermal anomalies and magmatism are linked to the tectonic evolution of the Apennine chain. Starting from the lower Miocene, the tectonic regime in the internal (western) part of the Apennines changed from compressional to extensional in response to the migration of the compressional front that moved eastward, and to the opening of the Tyrrhenian Sea, leading to crustal thinning, uplift of asthenospheric mantle, and magma generation. Indeed, a large part of the magmatism in Latium and Campania is of ultimate mantle origin, albeit modified by important evolutionary processes, whereas magmas with a crustal anatectic origin are present in Tuscany [132].

In Southern Tuscany, two geothermal areas (Larderello–Travale and Mt. Amiata) are presently exploited, and produce about 30% of the region's electricity demand. The Larderello–Travale field produces superheated steam from two reservoirs characterized by secondary permeability. The first, at a depth of 500–1500 m (T = 150–260 °C), is hosted in Mesozoic carbonate-evaporitic formations; the second, which is deeper (about 2.5–4 km depth) and hotter (T = 300–350 °C), is found inside a metamorphic Paleozoic basement,

in Plio–Quaternary acidic magmatic rocks intruding the basement, and in related thermo-metamorphic rocks [133–135]. The impermeable cover of the system consists of Tertiary flysch formations and Neogene sedimentary deposits. In spite of extensive literature on the Larderello field, data on Li the contents of fluids are comparatively scarce. The authors of [136] reported negligible Li content in the steam condensate ($\leq$0.2 mg/L); however, in some non-productive geothermal wells located in the peripheral parts of the field, they sampled and analyzed a liquid phase with Li concentrations up to 36.1 mg/L (Figure 8, Table S2).

The Mt. Amiata geothermal area includes two water-dominated systems (the Bagnore and Piancastagnaio fields). Additionally, in this case, similarly to the Larderello-Travale system, two geothermal reservoirs are exploited. They occur in Mesozoic carbonate-evaporitic rocks at about a 0.4–1.0 km depth (T = 150–260 °C), and in metamorphic Paleozoic rocks at about 2.5–3.5 km (T = 300–360 °C) [133]. The impermeable cover rocks are also similar to those of the Lardello–Travale area. The volcanic products of the Mt. Amiata volcano (0.3–0.2 Ma: [137] and references therein) generally overlie the impermeable cover rocks. The geothermal wells at Mt. Amiata have never encountered intrusive rocks, although the finding of thermo-metamorphic rocks in some deep wells suggests their occurrence. The two reservoirs produce a two-phase (liquid + vapour) mixture. The only available data on Li concentrations in the geothermal fluid at Mt Amiata refer to the brine resulting from flashing at the separator, and range from 3 to 35 mg/L (Figure 8, Table S2; [138–140]).

The eastern part of Latium is largely covered by Quaternary volcanic products, and it was the object of geothermal exploration until late 1950s. Several explorative geothermal wells were drilled in the areas of Mt. Vulsini, Vico–Cimini, and Sabatini, and the Albani Hills [141]. The exploration resulted in the identification of three main geothermal fields: Latera, Torre Alfina, and Cesano, and other less important geothermal occurrences. None of the investigated areas are presently exploited. The geothermal reservoir in all these areas occurs in Triassic–Oligocene formations, mainly consisting of carbonates with secondary permeability, whereas the impervious cover is made up mostly of Cretaceous–Miocene flysch units and, in places, of Neogene–Quaternary sediments or altered volcanic rocks [141–143]. At Latera, the carbonatic rocks in place are thermo-metamorphosed and intruded by a syenitic subvolcanic body and its related dykes [144].

In the Cesano geothermal area, twelve wells were drilled down to a depth of 3200 m. Two wells, located in the Baccano–Cesano Caldera, with temperatures of 210 °C and 150 °C, respectively, produced a super-saturated brine [141]. In particular, the brine from Cesano well 1 was characterized by a TDS of about 356,000 mg/L; moreover, it had a composition dominated by sodium, potassium, chloride, and sulphate, with very high concentrations of Li (380 mg/L), Rb (450 mg/L), and Cs (80 mg/L) [145]. According to these authors, the high contents of alkali metals in the brine can be related to an intense interaction between water and alkali-rich volcanic rocks of the RMP. As mentioned in the Introduction, this area is currently the object of a research permit [17].

Seven geothermal wells at Latera, studied in [143], produced a two-phase fluid with temperatures varying from 186 °C to 238 °C. The computed chemical composition of the reservoir fluids indicated that these fluids are Na-Cl waters with significant concentrations of carbonates and sulphates. The lithium contents of these fluids varied from 0.84 to 13.5 mg/L (Table S2). In addition, relatively high amounts of B (up to 576 mg/L), As (up to 106.2 mg/L), and Cs (up to 7.7 mg/L) were reported.

The only published data on Li concentration in the geothermal fluid of the Torre Alfina field are from a single analysis (4.7 mg/L) reported in [146], making reference to [147]. However, this last paper, apparently, does not contain Li data.

Additionally, in Campania, high-temperature-gradient zones (i.e., the Campi Flegrei caldera, west of Naples, and the Island of Ischia) are associated with volcanic centers [141]. In particular, during the early 1980s, 13 wells were drilled within the Campi Flegrei caldera: 9 in the western sector (Mofete area), and 4 in the central sector (San Vito area). Geothermal wells intercepted a sequence of volcanic and sedimentary rocks, including

pyroclastic products, tuffs, trachytic and latitic lavas, and marine sediments; the sequence
was hydrothermally altered and also affected by thermometamorphism in the deepest part
of the well (below a 1900–2000 m depth; [148]). The four wells, located in San Vito, were
characterized by modest permeability. Li contents of up to 16 mg/L were reported [149].
Of more interest are some of the wells in the Mofete zone, in which the authors found a
water-dominated system made up of three aquifers. The shallowest reservoir (550–1500 m
deep), made up of fractured volcanic rocks, was characterized by a fluid with TDS from
28,000 to 52,000 mg/L and temperatures from 230 to 308 °C. The other two reservoirs were
within thermometamorphosed sedimentary volcanic rocks. The intermediate reservoir
(1900 m deep) hosted a more diluted fluid (TDS = 18,200 mg/L) and had a temperature of
340 °C, whereas the deep reservoir (2700 m depth), with a temperature of 350 °C, contained
a hypersaline fluid (TDS = 200,000 mg/L) [141].

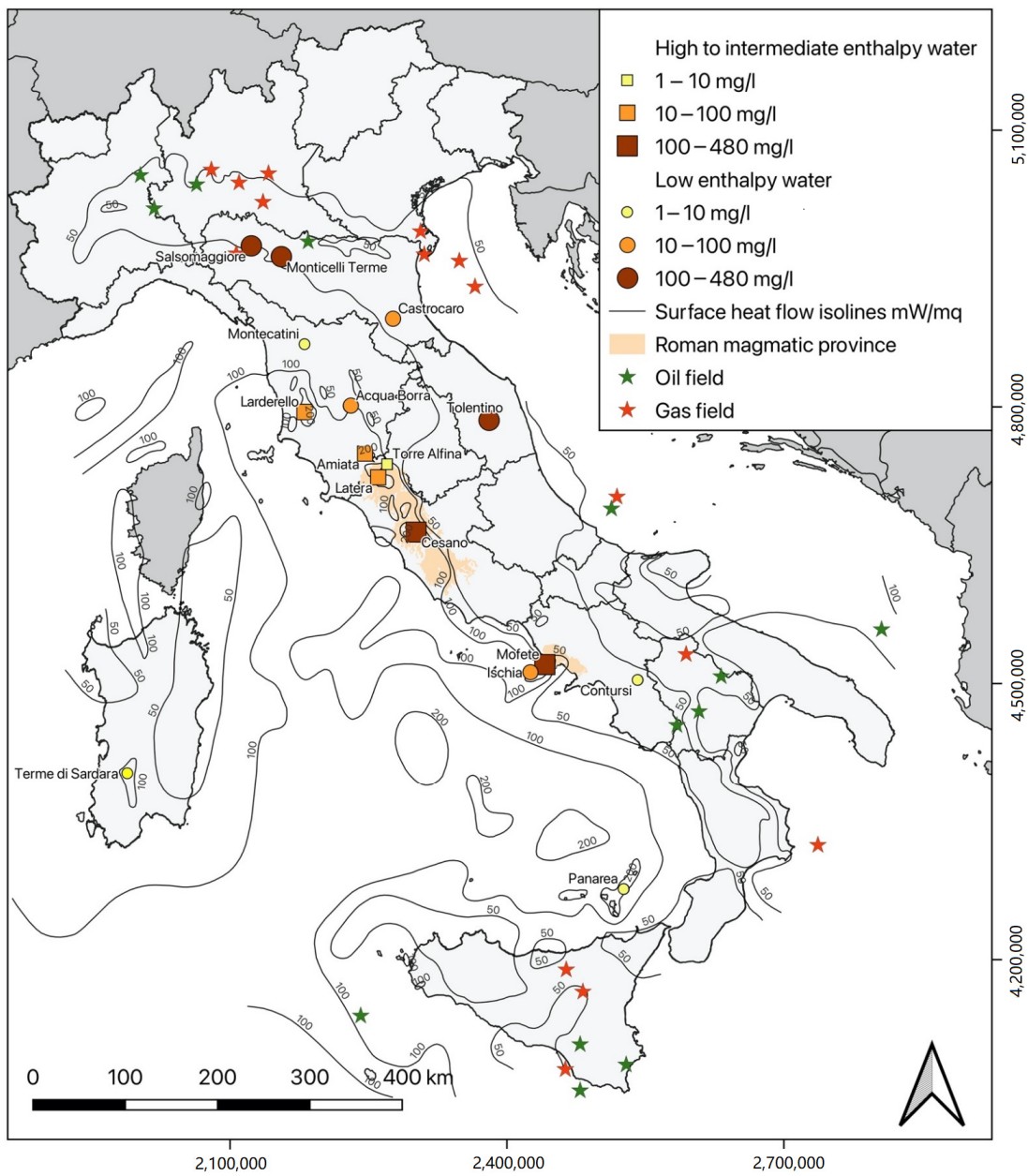

**Figure 8.** Location of Li-bearing fluids mentioned in text. Location of the oil and gas fields from [150].
Source of data: see Table S2. There are many other waters with Li contents of between 1 and
10 mg/L [20,151].

Lithium content is very high (480 mg/L) in the liquid phase produced by the deep reservoir, whereas is is lower, but still significant (28–56 mg/L), in the water discharged by the intermediate and shallow reservoirs (Figure 8, Table S2; [152]). The geochemical composition of Mofete geothermal fluid can be explained by the process of mixing between deep, uprising magmatic fluids with local meteoric waters and hot waters of marine origin [153]. None of the wells are currently exploited.

### 3.3.2. Low-Enthalpy

While high- to intermediate-enthalpy resources in Italy are associated with well-defined heat sources occurring at shallow levels in the crust, low-enthalpy resources can have different origins. Usually, low-enthalpy fluids occur in conduction-dominated geothermal systems, also called passive geothermal systems, where convective flows are negligible. Generally, these kinds of geothermal systems are typical of passive-plate margins, where no significant recent tectonism or volcanism occurs. These systems are usually located at a greater depth than convection-dominated systems, so faults can play an important role, influencing fluid pathways to act as either barriers or conduits [154].

Low-enthalpy thermal resources exist in several Italian regions, from Piedmont to Sicily [20]. In Tuscany, Latium, and Campania, low-enthalpy resources can exist either at the peripheral areas of the high- to intermediate-enthalpy geothermal fields, or because of a shallow fluid circulation. In the Northern Apennines, as well as in other Italian regions, low-enthalpy resources are recognized as a consequence of deep water circulation and upwellings through active non-sealed faults. Several of these waters have Li contents > 1 mg/L. Specifically, the database BDNG (Banca Dati Nazionale Geotermica—National Geothermal Database; [20]) lists as many as 60 thermal waters with Li ≥ 1 mg/L; a more recent compilation by [151] reports, for Southern Italy, nearly 40 occurrences with Li > 1 mg/L. In this review, we will focus on occurrences exceeding 10 mg/L (Table S2), although we will mention some springs with lower concentrations. Thermal waters exceeding 10 mg/L Li are mostly concentrated in the Emilia-Romagna, Campania, and Tuscany regions (Figure 8). The highest Li contents occur in the brines (TDS > 100 g/L) of the Northern Apennines foredeep, often associated with hydrocarbon reservoirs. The authors of [155] report an extreme value of 370 mg/L in the salty (TDS ~173 g/L) water at Tolentino (Figure 8) in the province of Macerata; in the same locality, four other water bodies display Li contents of 2.1–4.4 mg/L. At Salsomaggiore (Figure 8), in the province of Parma, [156] reports contents of up to 96.4 mg/L. The Supplementary File of the same paper reports earlier analyses in the same locality, indicating up to 121.5 mg/L, as well as up to 164 mg/L at Monticelli Terme (Parma), and 80 mg/L at Castrocaro (Forlì) (Figure 8). The high Li content in these brines is attributed to their interaction with sediments during early-to-late diagenesis and redox processes in the dolomitization occurring in the foredeep basin. The Mg–Li geothermometer applied to Salsomaggiore brines [156] gave a value of 156 ± 15 °C (mean ± SD; N = 12). Considering the present-day mean geothermal gradient of 30 °C/km, this temperature suggests a fluid circulation down to depths of at least 5 km.

Close to the southern slope of the Northern Apennines, in Montecatini town (Figure 8), high-salinity (TDS up to 17 g/L) thermo-mineral waters associated with a single, deep-fluid reservoir show Li concentrations of up to 5 mg/L [157,158]. The high TDS is attributed to the dissolution of the Triassic evaporitic layers, confirming the long-term deep circulation of water.

Several thermal water systems occur peripherally to the high-enthalpy Larderello–Travale and Mt. Amiata geothermal fields in Southern Tuscany [136,146,157,159]. In particular, in the Siena–Radicofani graben, there are at least three springs with Li concentrations close to 20 mg/L (Mortaione, Guado di Pietrafessa, and Acqua Borra; Figure 8). In these cases, lithium enrichment is ascribed to large convective circulation in highly permeable limestone reservoirs, which receive infiltration from meteoric waters and the likely leaking of saline waters from confined, self-sealed geothermal systems. Slightly

lower Li contents were recorded in another thermal spring peripheral to a geothermal field (Fonte Tiberio near Torre Alfina, 4.1 mg/L, [146]).

The authors of [160] studied the hydrogeochemistry of the Campania region in Southern Italy in order to relate the regional hydrogeology with the hydrothermal activity. In the Pozzuoli area at Terme di Nerone (also known as Stufe di Nerone) Li concentrations up to 10.1 mg/L were revealed. The application of several geothermometers for this spring gave inconsistent computed temperatures (11–343 °C); the authors believe that these fluids are not related to the Campi Flegrei high-temperature geothermal system, but rather, they have a shallow origin. The authors of [161] further document remarkable Li contents in the thermal springs of Campi Flegrei (up to 10 mg/L) and of the closely related Ischia Island (Figure 8) (up to 6 mg/L). For the same Ischia Island thermal waters, [162] reports up to 41 mg/L of Li. In Campania, Contursi springs 2 and 3 are also noteworthy, and present an amount of Li in their waters of 8.74 mg/L [160]. Contursi is a non-volcanic area in the Campania Apennines; these waters are supposed to circulate deep in the carbonate units, and are the result of a fast ascent along the main active faults bordering the carbonate formations. Other systems where deep circulation along faults results in Li-rich thermal waters occur along the Sardinia rift and the Campidano graben, Sardinia [163]. There are several occurrences of Li contents up to 7.7 mg/L (Table S2).

Finally, we can mention a study devoted to the definition of a geochemical conceptual model of thermal fluids, and to the establishment of possible relations of deep-origin fluids with the magmatic system of Panarea Island [164]. In this context, spring Panarea 5 presented a Li concentration of 7.5 mg/L. The relatively high values of minor elements such as Li can be attributed to a mixing process involving sea water and hot, low-pH, Cl-rich end-members.

## 4. Discussion and Conclusions

From the previous description, it appears that Italy has a modest potential for conventional hard-rock Li deposits. A few spodumene pegmatites crop out only in Alto Adige (e.g., Val Racines; Figures 2 and 3), at the western end of the relevant Austrian spodumene belt. There are, however, a number of contexts in which unconventional hard-rock ores require thorough studies. Granites, pegmatites, and/or greisens containing lithium micas (e.g., zinnwaldite, lepidolite) are becoming an interesting target thanks to the recent advances in industrial processes, e.g., [165,166]. Intrusive rocks from TMP provide some examples; however, the San Piero LCT pegmatites have a negligible bulk tonnage, whereas the Capo Bianco Aplite lies in a spectacular location in the Tuscan Archipelago National Park, and any exploitation does not appear feasible. They must be considered only as natural laboratories for scientific studies. On the other hand, the Late Variscan intrusive rocks in Sardinia, especially the peraluminous F-rich systems in the south, and the Calabrian batholiths should be investigated in more detail. A second unconventional target could be represented by the post-Variscan, Permian volcano-sedimentary sequences of Northern Italy. Here, sedimentation and weathering occurred together with concomitant volcanic and hydrothermal activity, providing the right conditions for lithium leaching and concentration in Jadar-like deposits. The Val Tanaro tainiolite–danburite occurrence could be an example of such deposits, although they were metamorphically reworked during the Alpine event.

As already stated in the previous sections, Italy does not host conventional salar deposits. On the other hand, the distinct Li-rich characteristics of recent (Plio–Quaternary) magmatic rocks of the Tuscan and Roman magmatic provinces suggests that there may be other potential unrecognized, unconventional fluid resources. Indeed, this Li-rich nature of magmatic rocks seems to be reflected in the occurrence of Li-rich geothermal fluids (although the Li in these fluids is not necessarily of magmatic origin). Specifically, several high- to intermediate-enthalpy fluids present Li concentrations in the range of 10–500 mg/L Figure 9). Moreover, nearly all these fluids have very low Mg contents, and therefore, very low Mg/Li ratios (mostly < 1; Table S2; Figure 9). The separation of Mg

and Li from brines is challenging because of the quite similar chemical behavior of the two elements. Therefore, the Mg/Li ratio is a critical parameter that determines the economic feasibility of Li recovery from fluids; the optimal ratios are typically < 10 ([166,167] and references therein). Despite occurring in different geological contexts, a common feature of Italian high-enthalpy fluids is that they originated from/interacted with Li-rich, Mg-poor source(s), and/or were able to develop higher Li concentration than Mg very efficiently. At present, we have no conclusive evidence of the source(s) and process(es) involved; however, we notice that a recent study [168] indicates that aqueous solutions interacting with a granitic rock at T > 200 °C result in fluids with low Mg/Li ratios. Whatever the process(es) involved, this concurrence of high Li contents and low Mg/Li ratios makes geothermal fluids the first potential objective for Italian Li sources, as demonstrated by the existence of a research permit in the Cesano area [17]. Moreover, based on the above reported experimental evidence [168], one can envisage a visionary perspective whereby a deep injection of water is implemented to interact with, and extract Li from, magmatic rocks. However, the exploitation of deep fluids may face problems with social acceptance. Social acceptance has become a key issue in the development of any activity, including mining, e.g., [169]. With specific reference to lithium, there are several social issues connected with salar exploitation in South America [170], and the development of the Jadar deposit met significant opposition [171,172].

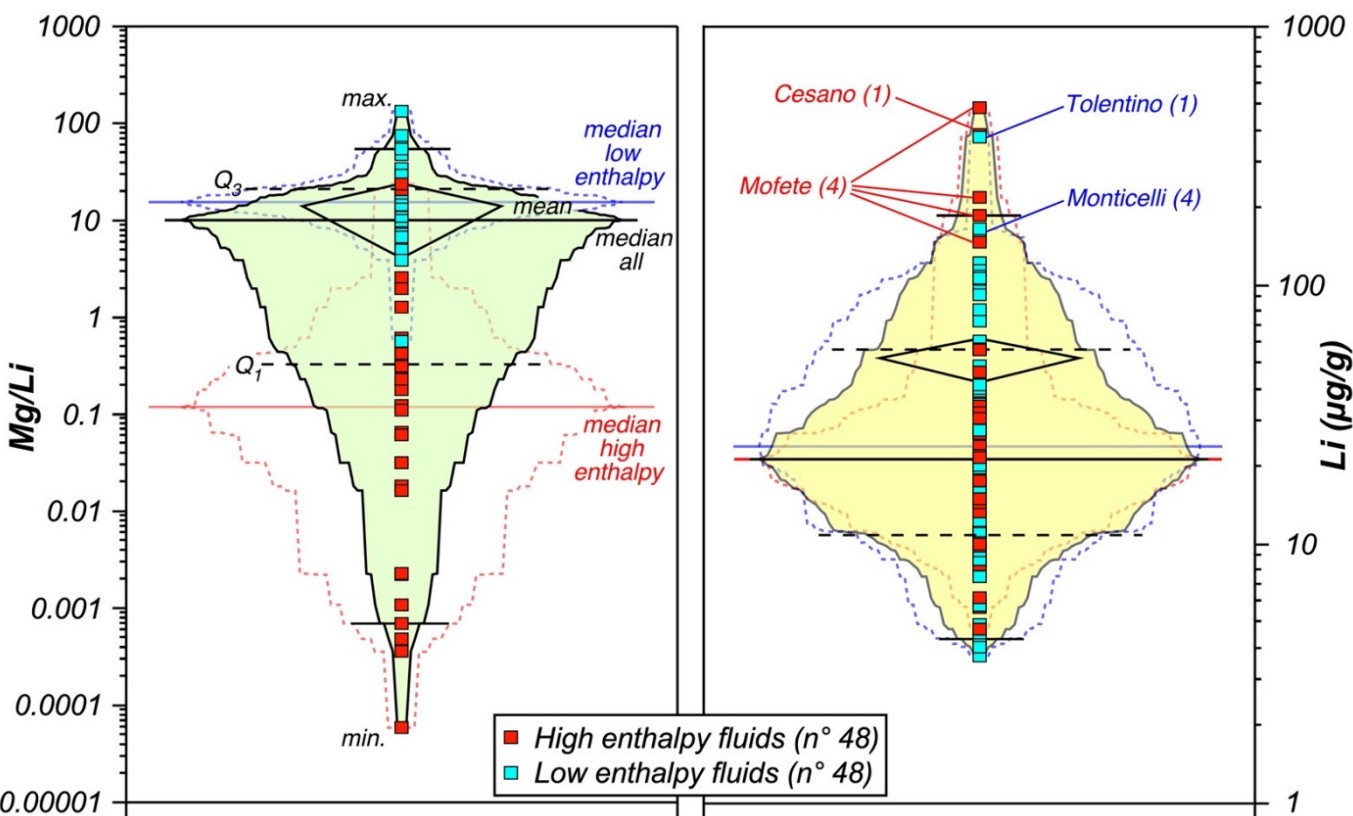

**Figure 9.** Box-percentile plots of Mg/Li ratios and Li contents of Italian thermal waters. Source of data: see Table S2.

In Italy, several groups and local authorities oppose deep drilling to exploit intermediate- to high-enthalpy fluids, e.g., [173]. Therefore, any strategy for the exploitation of geothermal lithium should include adequate steps to achieve social acceptance.

Low-enthalpy fluids have typically lower Li contents, but may be more socially acceptable, particularly thermal springs (e.g., if associated with the development of spa facilities). These waters also have quite variable Mg contents (Table S2) and typically higher Mg/Li ratios, reflecting an origin of/interactions with heterogenous sources depending on the

specific location (for instance, the high Mg content and high Mg/Li ratio at Panarea are clearly the result of the involvement of marine water [164]. However, with few exceptions, most of the described occurrences show Mg/Li ratios between 4 and 30 (Table S2, Figure 9), i.e., still compatible with the current separation technologies [166,167].

In summary, there are a few contexts in Italy that may warrant a systematic investigation of hard-rock Li deposits, especially considering the current geo-political situation. Geothermal fluids hold better perspectives, especially because they present quite favorable Mg/Li ratios. This type of resource seems to be the most credible short-term target, and should be implemented through detailed studies of the specific settings and origin(s) of the fluids. Moreover, exploration and exploitation should be supported by measures directed toward social acceptance. Mid-term strategies should include the evaluation of unconventional resources such as low-enthalpy waters, and the direct leaching of Li-rich magmatic rocks.

**Supplementary Materials:** The following supporting information can be downloaded at: https://www.mdpi.com/article/10.3390/min12080945/s1, Table S1: "Hard-rock" Li occurrences in Italy; Table S2: Selected Li-bearing fluids in Italy.

**Author Contributions:** Conceptualization, A.D. and P.L.; literature search, A.D., P.L., G.R. and E.T.; writing—original draft preparation, P.L.; writing—review and editing, A.D., P.L., G.R. and E.T.; visualization, A.D. and E.T. All authors have read and agreed to the published version of the manuscript.

**Funding:** This research received no external funding.

**Data Availability Statement:** All data and information used for this review are contained in the references listed and in the Supplementary Tables.

**Acknowledgments:** This study was entirely conducted in the absence of specific funding. Nonetheless, we thank the CNR Institute of Geosciences and Earth Resources (IGG) for providing office space to A.D., G.R. and E.T., and for providing bibliographic resources. Adele Manzella steadily encouraged us to carry on with the study; Patrizia Fiannacca, Marilena Moroni, and Stefano Naitza kindly discussed some specific topics; and Mirko and Lodovico Grisotto, Roberto Appiani, Cristian Biagioni, and Marco Lorenzoni supplied information and photographic images.

**Conflicts of Interest:** The authors declare no conflict of interest.

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
