# Peer review of "Lithium Occurrence in Italy—An Overview"

_minerals, doi:10.3390/min12080945_

Round 1
Reviewer 1 Report
Well-designed synthesis work with an extensive bibliographic framework, both in the Italian region and globally.
Recommendation: In Figure 6 replace scistosity by foliation
Reviewer 2 Report
This is an interesting paper that reviews the perspective for Li resource in Italy, including the known lithium occurrences in Italy, with an emphasis on the potential recovery from geothermal fluids. It is suitable for publication after a minor modification. The revised paper should address the following questions:
1. Line 23-24, you noted ‘Moreover, a visionary, but not impossible, perspective may consider deep injection of water to interact with, and extract Li from, magmatic rocks. The conclusion needs evidences enough.
2. In figure 2, Tertiary-Quaternary magmatic rocks should be added. In the abstract, lines 15-17, you wrote ‘we notice that Tertiary-Quaternary magmatic rocks in Tuscany and Latium have Li contents systematically higher than recorded in normal arc igneous rocks worldwide’. However, I could not find the spatial distribution of Tertiary-Quaternary magmatic rocks.
3. Line 142, you noted ‘the Tertiary magmatism (Figure 2)’, but I cannot find the Tertiary magmatism in Figure 2.
4. Line 698-699, you noted ‘Moreover, nearly all these fluids have very low Mg contents, and therefore Mg/Li ratios << 10 (Table S2; Figure 9), which is considered optimal for Li recovery from fluids’. You should further illustrate the relationship between Mg/Li and Li recovery from fluids, and much more geological or geochemical evidences should be shown in the text.
5. Line 715, figure 9, What’s the meaning of this figure? If trying to describe the relationship between Li and Mg/Li, in the fluid, you can show clearly it using regression analysis instead of the boxplot.
6. In figure 1, a scale should be added.
7. Line 83, ‘Geological context’ should be changed into ‘Geological background’.
8. Line 729, ‘Italy seems to have a low potential for hard rock Li deposits, because of a largely unfavorable geology;’ can be deleted, which is not full evidences enough, more research works are required.
